# Multi-Objective Intrinsic Reward Learning for Conversational Recommender Systems

**Zhendong Chu**
University of Virginia
zc9uy@virginia.edu
Charlottesville, VA, USA

**Nan Wang**[*]
Netflix Inc.
nanw@netflix.com
Los Gatos, CA, USA

**Hongning Wang**
University of Virginia
hw5x@virginia.edu
Charlottesville, VA, USA

## Abstract

Conversational Recommender Systems (CRS) actively elicit user preferences to generate adaptive recommendations. Mainstream reinforcement learning-based CRS solutions heavily rely on handcrafted reward functions, which may not be aligned with user intent in CRS tasks. Therefore, the design of task-specific rewards is critical to facilitate CRS policy learning, which remains largely under-explored in the literature. In this work, we propose a novel approach to address this challenge by learning intrinsic rewards from interactions with users. Specifically, we formulate intrinsic reward learning as a *multi-objective bi-level optimization problem*. The inner level optimizes the CRS policy augmented by the learned intrinsic rewards, while the outer level drives the intrinsic rewards to optimize two CRS-specific objectives: *maximizing the success rate* and *minimizing the number of turns to reach a successful recommendation* in conversations. To evaluate the effectiveness of our approach, we conduct extensive experiments on three public CRS benchmarks. The results show that our algorithm significantly improves CRS performance by exploiting informative learned intrinsic rewards.

## 1 Introduction

Conversational recommender systems (CRS) leverage interactive conversations to delineate a user's preferences [Zhang et al., 2018, Lei et al., 2020a, Chu et al., 2023]. The conversations revolve around questions aimed at discerning users' preferences on specific item attributes (e.g., music genres). Through an interactive process of questions and answers, a profile about a user's intended item can be depicted. Numerous CRS formulations have been proposed [Chen et al., 2019, Christakopoulou et al., 2018, 2016]. In this work, we investigate a prevalent CRS setting known as the multi-round conversational recommendation [Lei et al., 2020a, Deng et al., 2021], where a CRS agent can ask a question or recommend an item in consecutive rounds of conversations. The conversation continues until the user accepts the recommendation (indicating a successful conversation) or quits the conversation (considered as a failed conversation).

CRS fundamentally embodies a sequential decision making problem, for which numerous reinforcement learning (RL)-based solutions have been proposed [Lei et al., 2020a, Chu et al., 2023]. However, as the users only provide textual or binary responses (e.g., accepting or rejecting the inquired attributes), existing RL-based solutions heavily rely on heuristic reward functions that are manually defined to train CRS policies. These reward functions, such as promoting attributes accepted by a user and penalizing those rejected, may not accurately reflect user intent due to their heuristic nature. This becomes problematic since the effectiveness of CRS policy learning largely depends on the quality of pre-defined reward function – an inadequately designed reward function can lead to solutions that

---

[*]Work was done while at the University of Virginia.

37th Conference on Neural Information Processing Systems (NeurIPS 2023).

significant deviates from optimality. Additionally, these arbitrary reward functions can inadvertently distort the modeling of conversation states, influencing the subsequent actions taken by the RL agent.

Arguably, an effective reward function should promote actions that lead to more precise modeling of users' preferences. As a result, different attributes or items, including those rejected, can each uniquely contribute to user preference modeling. As an example illustrated in Figure 1, even though *Heavy metal rock* is rejected by the user, it still, to certain extent, contributes to identifying the target item, *Hey Jude*. However, existing handcrafted heuristic reward functions fall short in delivering information at this granularity, as they assign uniform rewards to all accepted or rejected actions. This motivates us to *learn a reward function* that enables more fine-grained policy learning.

Instead of manually define reward functions, we introduce a principled approach to reward learning for CRS, where we learn a *intrinsic reward* for each action taken by the agent utilizing the optimal rewards framework [Singh et al., 2010]. This framework delineates the optimal intrinsic reward function as the one that, when employed by an RL agent, fosters behaviors that optimize the task-specific or *extrinsic rewards* – in the case of CRS, successful recommendations.

Two notable technical challenges stand out when learning intrinsic rewards for CRS. First, explicit extrinsic rewards in CRS are extremely sparse, which com-

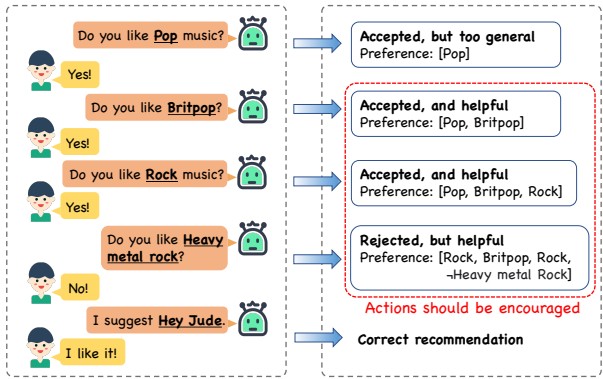

Figure 1: Motivating example of intrinsic reward learning.

plicates the intrinsic reward learning. Despite that the agent interacts with the user in each round, the only clear extrinsic reward signal, which is whether the overall conversation is successful or not, is only revealed from the user at the conclusion of the conversation. The significance of each accepted or rejected attribute/item prior to the conversation's ends remain ambiguous. For instance, an inquired attribute that is rejected by the user does not necessarily imply a negative reward for policy learning, as it can signify what the user is not looking for. Second, the assessment of CRS is multi-dimensional, entailing various factors that contribute to the overall effectiveness and user experience, such as recommendation quality and user effort. On the one hand, asking more questions may be necessary to accurately profile user preferences to facilitate a successful recommendation. On the other hand, reducing user effort in conversations (i.e., fewer conversation turns) is essential to ensure users' engagement and maintain their satisfaction. Balancing these factors is a delicate task.

To tackle the challenges for improving CRS from a reward learning perspective, we develop an online algorithm for learning intrinsic reward functions via multi-objective bi-level optimization. We name the proposed solution **CRSIRL**, meaning **CRS** with **I**ntrinsic **R**eward **L**earning. In the inner loop of CRSIRL, the policy is optimized with the learned intrinsic reward function. In the outer loop, the intrinsic reward function is updated to satisfy two specific criteria designed for CRS. The first criterion aims to maximize the sparse extrinsic reward, augmented by a reward shaping strategy to encourage actions that promote the target item as quickly as possible. The second criterion involves tailoring the learnt reward function to promote successful trajectories over the failed ones. The results of our extensive experiments demonstrate that CRSIRL not only improves the success rate of CRS but also achieves it with shorter conversations.

## 2 Related Work

**Conversational Recommder Systems.** Christakopoulou et al. [2016] pioneered the concept of Conversational Recommender Systems (CRS). Their approach primarily focused on determining which items to solicit feedback on and applied off-the-shelf metrics such as the upper confidence bound [Auer, 2002] for this purpose. This laid the groundwork for reinforcement learning (RL) based methods, which have recently become the prevalent solutions for CRS. For example, Sun and Zhang [2018] developed a policy network to decide whether to recommend an item or inquire about an item attribute at each conversation turn. However, these initial studies terminated the conversation upon

making a recommendation, regardless of user acceptance. Lei et al. [2020a] studied multi-round conversational recommendation, where CRS can ask a question or recommend an item in multiple rounds before the user accepts the recommendation or quits. This is also the setting of our work in this paper. To better address multi-round CRS, Lei et al. [2020b] leveraged knowledge graphs to select more relevant attributes to ask across turns. Xu et al. [2021] extended [Lei et al., 2020a] by revising user embeddings dynamically based on users' feedback on attributes and items. And Deng et al. [2021] unified the question selection module and the recommendation module in an RL-based CRS solution. However, all the aforementioned works depend on heuristically crafted reward functions, which may lead policies to deviate from the optimal solution. In this work, we propose to learn intrinsic rewards which can maximize the recommendation performance.

**Intrinsic Reward Learning in Reinforcement Learning.** Intrinsic reward learning has emerged as a promising approach to enhance the performance and efficiency of reinforcement learning algorithms. Singh et al. [2010] introduced the Optimal Reward Framework which aims to find a good reward function that allows agents to solve a distribution of tasks using exhaustive search. Pathak et al. [2017] introduced the concept of curiosity-driven intrinsic rewards, where the agent is rewarded for actions that lead to novel states, improving its ability to explore complex environments. Zheng et al. [2018] proposed a meta-gradient method named LIRPG to learn intrinsic rewards via a bi-level optimization framework. Zheng et al. [2020] extended LIRPG by learning intrinsic rewards on a distribution of tasks. Liu et al. [2022] developed another meta-gradient method to learn intrinsic rewards from trajectory preferences. In this work, we propose a novel intrinsic reward learning framework designed for CRS, where we learn intrinsic reward functions to satisfy multiple CRS-sepcific objectives from users' extremely sparse explicit reward feedback.

## 3 Preliminaries

In this section, we define the notations to be used in our technical discussions and some basic notions in multi-objective optimization.

### 3.1 Problem Definition

Similar to traditional recommender systems, CRS serves a set of users $\mathcal{U}$ with a set of items $\mathcal{V}$; and we denote a specific user as $u$ and an item as $v$. Each item $v$ is associated with a set of pre-defined attributes $\mathcal{P}_v$. Attributes describe basic properties of the items, such as genres in movie recommendations and cuisine type in restaurant recommendations.

We formulate the CRS problem using a Markov decision process (MDP) [Deng et al., 2021, Lei et al., 2020b, Chu and Wang, 2023], which can be fully described by a tuple $(\mathcal{S}, \mathcal{A}, \mathcal{T}, \mathcal{R})$. $S$ denotes the state space, which summarizes the conversation between the system and user so far. $\mathcal{A}$ denotes the action space for the system, which includes recommending a particular item or asking for feedback on a specific attribute. $\mathcal{T} : \mathcal{S} \times \mathcal{A} \to \mathcal{S}$ is the state transition function, and $\mathcal{R} : \mathcal{S} \times \mathcal{A} \to \mathbb{R}$ is a reward function.

With this formulation, a conversation in CRS can be represented as $d = \{(a_1, r_1), ...(a_T, r_T)\}$, where $T$ is the maximum number of allowed turns. A conversation (or an episode in the language of RL, which we will use exchangeably) terminates when: (1) the user accepts the recommended item; or (2) the CRS agent runs out of maximum number of allowed turns. At each time step $t$, the CRS agent, which follows a policy $\pi_\theta(a_t|s_t)$ parameterized by $\theta$, selects an action $a_t$ based on the current state $s_t$. The training objective of a CRS policy is to maximize the expected cumulative rewards over the set of observed episodes $\mathcal{D}$, i.e., minimizing the loss

$$\mathcal{L}(\pi) = - \mathop{\mathbb{E}}_{d \sim P(\mathcal{D})} \Big[ \sum_{t=0}^{T} R_t \Big], \tag{1}$$

where $R_t = \sum_{t'=t}^{T} \gamma^{T-t'} r(a_t)$ is the accumulated reward from turn $t$ to the final turn $T$, and $\gamma \in [0, 1]$ is a discount factor to emphasize rewards collected in a near term.

Instead of using handcrafted reward functions $\mathcal{R}$ as in previous works [Lei et al., 2020a, Deng et al., 2021, Chu et al., 2023], we learn an intrinsic reward function defined as $r_\phi^{in}(s, a)$ parameterized by $\phi$ to enhance CRS policy learning. In this context, the original CRS-specific reward is referred to

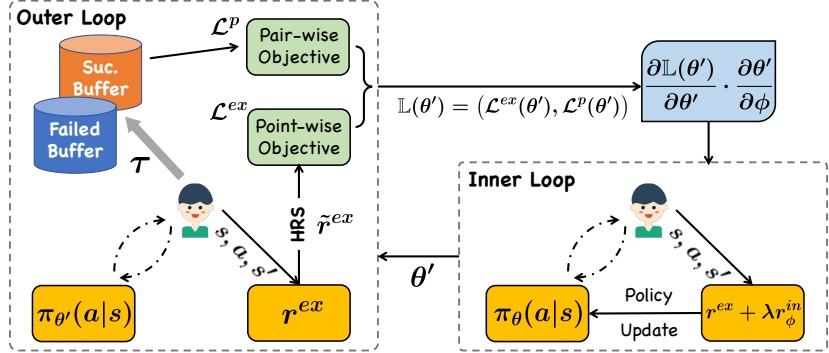

Figure 2: Overview of CRSIRL,which consists of two modules, a policy parameterized by $\theta$ and an intrinsic reward function parameterized by $\phi$. The optimization of CRSIRL has two levels. In the inner level, a policy is trained to maximize the return defined by both intrinsic and extrinsic rewards. In the outer level, the intrinsic reward function is trained to optimize two CRS-specific objectives realized by the learnt policy's behaviors.

as the extrinsic reward, denoted as $r^{ex}(s, a)$. The extrinsic reward is inherently sparse, as the only discernible and useful reward signal is the success or failure of an episode, with the intermediate actions' contributions remaining ambiguous. We assign a positive extrinsic reward at the conclusion of a successful episode and a negative reward otherwise. All intermediate actions are assigned a zero extrinsic reward.

### 3.2   Multi-Objective Optimization

We utilize multi-objective optimization (MOO) to achieve the multi-dimensional goal of CRS, i.e., maximizing the success rate and reducing the length of conversations. MOO aims to simultaneously optimize multiple objectives, possibly conflicting ones. This results in a trade-off among objectives, making the CRS problem more complex and challenging to solve. In these cases, the Pareto-optimal solutions represent different optimal trade-offs between the objectives [Deb and Deb, 2013].

Consider $M$ objective functions $\{\mathcal{L}^1, ..., \mathcal{L}^M\}$, a model parameterized by $\theta$ is optimized towards them. We specify the multi-objective optimization using a vector valued loss $\mathbb{L}$,

$$\min_{\theta} \mathbb{L}(\theta) = \min_{\theta} \left( \mathcal{L}^1(\theta), ..., \mathcal{L}^M(\theta) \right)^{\top} \tag{2}$$

The goal of multi-objective optimization is achieving Pareto optimality.

**Definition 1** (Pareto optimality).

   (a) A solution $\theta$ dominates a solution $\bar{\theta}$ if $\mathcal{L}^t(\theta) \leq \mathcal{L}^t(\bar{\theta})$ under all objectives and $\mathbb{L}(\theta) \neq \mathbb{L}(\bar{\theta})$.

   (b) A solution $\theta^*$ is called Pareto optimal if there exists no solution $\theta$ that dominates $\theta^*$.

The set of Pareto optimal solutions is called the Pareto front $\mathcal{P}_{\theta}$.

## 4   Multi-Objective Intrinsic Reward Learning for CRS

In this section, we elaborate on the structure of CRSIRL, as illustrated in Figure 2. CRSIRL operates on two tiers of optimization: the inner optimization, which improves the policy using both the extrinsic reward and the learned intrinsic reward function, and the outer optimization, which refines the intrinsic reward function based on the policy assessment derived from the inner optimization. Given the absence of supervision for the intrinsic reward function, we establish the relationship between it and the refined policy via gradient descent in the inner optimization. Specifically, we compute meta-gradient for the intrinsic reward function using chain rule in the outer optimization. In the outer optimization, we design a point-wise objective striving to enhance extrinsic rewards in the learnt intrinsic reward function, through the use of hindsight reward shaping (HRS). This objective aids in pinpointing pivotal actions that significantly improve the target item's ranking, thereby shortening the conversation length. In parallel, we introduce a pair-wise objective that favors successful trajectories over unsuccessful ones, which assists in identifying actions that result in

preferred conversations. This objective is named as recommendation preference matching (RPM). Finally, we introduce a holistic multi-objective bi-level optimization framework that optimizes intrinsic rewards to meet both objectives.

## 4.1 Hindsight Reward Shaping

As we discussed before, the extrinsic reward is extremely sparse in CRS. The only clear and informative signal from the extrinsic reward is whether the conversation is successful, making it hard to judge the progress of user preference elicitation during the conversation. Reward shaping, as proposed by Ng et al. [1999], serves as a valuable tool for incorporating task-specific knowledge to estimate the reward function. We leverage reward shaping within the outer loop of our model to imbue the process of intrinsic reward learning with more nuanced, task-specific guidance. We use the following hindsight reward shaping to augment the extrinsic reward,

$$\tilde{r}^{ex}(s_t, a_t) = r^{ex}(s_t, a_t) + \gamma w(s_{t+1}, v) - w(s_t, v), \tag{3}$$

where $w$ is a scoring function, $v$ is the target item and $\gamma$ is a discount factor. $\tilde{r}^{ex}(s_t, a_t)$ encourages actions which promote the target item. In turn, it helps shorten the conversation length. In our experiments, we use $w = \log(\rho(s_t, v) + 1)$, where $\rho(s_t, v)$ is the rank of target item $v$ under state $s_t$.

**Lemma 1.** *Consider any reward shaping function $\mathcal{F} : \mathcal{S} \times \mathcal{A} \times \mathcal{S} \to \mathbb{R}$, we say $\mathcal{F}$ is a **potential-based reward shaping function (PBRS)** if there exists a real-valued function $\Phi : \mathcal{S} \to \mathbb{R}$ satisfying,*

$$\mathcal{F}(s, a, s') = \gamma \Phi(s') - \Phi(s), \tag{4}$$

*Then $\mathcal{F}$ being PBRS is a necessary and sufficient condition for it to guarantee the consistency of the optimal policy, i.e., the optimal policy of $(\mathcal{S}, \mathcal{A}, \mathcal{T}, \mathcal{R}+\mathcal{F})$ is the same as $(\mathcal{S}, \mathcal{A}, \mathcal{T}, \mathcal{R})$.*

The proof is based on [Ng et al., 1999] and omitted in this paper. By matching the form of Eq.(4) and Eq.(3), we can conclude the hindsight reward shaping satisfies the PBRS condition, and thus the optimal policy is consistent. The resulted objective induced by HRS is

$$\mathcal{L}^{ex}(\theta) = -\mathbb{E}\Big[\sum_{t=0}^{T} \tilde{R}_t^{ex}\Big], \tag{5}$$

where $\tilde{R}_t^{ex} = \sum_{t'=t}^{T} \gamma^{T-t'} \tilde{r}_{t'}^{ex}$. Note that the information of target item is unknown beforehand, we can only use HRS after the target item is hit, which is why we call it *hindsight*. Otherwise HRS degenerates to the original extrinsic reward.

## 4.2 Recommendation Preference Matching

Even though the contributions of intermediate actions to a conversation are undefined in the extrinsic reward, it is still feasible to discern valuable intermediate actions that could potentially lead to a successful conversation, and the learned intrinsic reward should help us identify them. We realize this by contrasting successful and failed episodes by the learnt intrinsic reward: a preferred episode should have a higher likelihood under the optimal policy, comparing to a less preferred one; and this optimal policy should be achieved by the correct reward function. Given a policy $\pi_\theta$, the probability of conversation $\tau^0$ is preferred over $\tau^1$ is computed based on the likelihood of the trajectories,

$$P_\theta\big[\tau^0 \succ \tau^1\big] = \frac{\exp \sum_{t \in \tau^0} \log \pi_\theta(a_t|s_t)}{\exp \sum_{t \in \tau^0} \log \pi_\theta(a_t|s_t) + \exp \sum_{t \in \tau^1} \log \pi_\theta(a_t|s_t)}, \tag{6}$$

Assume $\tau^0$ is preferred over $\tau^1$, the resulting loss function is given by,

$$\mathcal{L}^p(\theta) = -\mathbb{E}\Big[\sum_{\tau^0 \succ \tau^1} \log P_\theta\big[\tau^0 \succ \tau^1\big]\Big], \tag{7}$$

where $\tau^0, \tau^1 \in \mathcal{B}$ are sampled from a buffer storing past trajectories. This follows the Bradley-Terry model [Bradley and Terry, 1952] for estimating score functions from pairwise preferences. In the context of CRS, the preference is naturally defined by whether the recommendation is successful or not; and among successful recommendations, we prefer the one shorter. We truncate the failed trajectory to match the length of the successful trajectory.

## 4.3 Multi-Objective Bi-Level Optimization

The intrinsic reward function is expected to lead to a policy satisfying the above two objectives. This translates to a bi-level optimization procedure for policy learning: first update the policy with learned intrinsic rewards, and then improve the intrinsic rewards to help the resulting policy better satisfy the above two objectives. More formally, we define,

$$
\begin{aligned}
&\min_{\phi} \ \mathbb{L}(\theta'), \\
&\text{s.t.} \ \ \theta' = \arg\min_{\theta} \mathcal{L}^{ex+in}(\theta, \phi).
\end{aligned}
\tag{8}
$$

where $\mathbb{L}(\theta') = \left(\mathcal{L}^{ex}(\theta'), \mathcal{L}^{p}(\theta')\right)$ and $\mathcal{L}^{ex+in}(\theta, \phi)$ is the negative cumulative reward calculated with weighted sum $r^{ex} + \lambda r^{in}_{\phi}$, $\lambda$ is a hyper-parameter to balance two rewards. In the inner loop, we optimize the policy with both the extrinsic reward and the learned intrinsic reward function. In the outer loop, we optimize the intrinsic reward function to minimize the vector value loss. To derive the gradients for optimization, we first build the connection between $\theta$ and $\phi$ in the inner loop, and then derive the gradients on $\phi$ in the outer loop.

**Inner Loop: Optimizing $\theta$, building the connection between $\theta$ and $\phi$.** We update $\theta$ as follows,

$$
\theta' = \theta - \eta \cdot \nabla_{\theta} \mathcal{L}^{ex+in}(\theta, \phi),
\tag{9}
$$

where $\nabla_{\theta} \mathcal{L}^{ex+in}(\theta, \phi)$ can be calculated by the policy gradient theorem [Sutton et al., 1999] and $\eta$ is the learning rate used in the inner loop. In this way, the updated parameter $\theta'$ becomes a function of $\phi$. With the built connection, we are able to compute the gradient of $\phi$ by taking the gradient of gradient on $\theta'$, i.e., the *meta-gradient*.

**Outer Loop: Optimizing $\phi$.** In the outer loop, we optimize the vector value loss $\mathbb{L}(\theta')$ to satisfy aforementioned two CRS-specific objectives. Even though we do not have supervision on $\phi$, the gradient of $\phi$ can still be derived using the chain rule,

$$
g(\phi) = \frac{\partial \mathbb{L}(\theta')}{\partial \theta'} \cdot \frac{\partial \theta'}{\partial \phi}
\tag{10}
$$

Different from single objective optimization, the first part of Eq.(13) is the derivative w.r.t. the multi-objective function $\mathbb{L}(\theta')$. Sener and Koltun [2018] adopt the multiple gradient descent algorithm (MDGA) [Désidéri, 2012] to find a Pareto stationary point for a MOO problem. We follow their approach to solve the following optimization problem,

$$
\min_{\alpha \cdot \in [0,1]} \left\{ \left\| \alpha \nabla_{\theta'} \mathcal{L}^{ex}(\theta') + (1 - \alpha) \cdot \nabla_{\theta'} \mathcal{L}^{p}(\theta') \right\|_{2}^{2} \right\},
\tag{11}
$$

where $\alpha$ has the following analytical solution,

$$
\alpha = \left[ \frac{\nabla_{\theta'} \mathcal{L}^{p}(\theta') - \nabla_{\theta'} \mathcal{L}^{ex}(\theta')^{\top} \nabla_{\theta'} \mathcal{L}^{p}(\theta')}{\left\| \nabla_{\theta'} \mathcal{L}^{ex}(\theta') - \nabla_{\theta'} \mathcal{L}^{p}(\theta') \right\|} \right]_{+,\frac{1}{\top}},
\tag{12}
$$

where $[\cdot]_{+,\frac{1}{\top}}$ represents clipping to $[0,1]$ as $[a]_{+,\frac{1}{\top}} = \max(\min(a, 1), 0)$. The resulted meta-gradient of $\phi$ becomes,

$$
g(\phi) = \alpha \cdot \frac{\partial \mathcal{L}^{ex}(\theta')}{\partial \theta'} \cdot \frac{\partial \theta'}{\partial \phi} + (1 - \alpha) \cdot \frac{\partial \mathcal{L}^{p}(\theta')}{\partial \theta'} \cdot \frac{\partial \theta'}{\partial \phi}.
\tag{13}
$$

Thus $\phi$ is updated by,

$$
\phi' = \phi - \beta \cdot g(\phi),
\tag{14}
$$

where $\beta$ is the learning rate used in the outer loop. We can conclude the optimization in the outer loop as an automatic trade-off between two objectives, and thus the resulted intrinsic reward function is expected to strike a good balance between two CRS objectives.

**Training procedure.** Now we are finally equipped to illustrate the complete learning solution for CRSIRL in Algorithm 1. In the inner loop, we first rollout an episode to calculate $\mathcal{L}^{ex+in}$. In the outer loop, we also rollout an episode to calculate $\mathcal{L}^{ex}$ and sample a pair from the trajectory buffer $\mathcal{B}$ to calculate $\mathcal{L}^{p}$. We run the inner loop and outer loop alternately until the model convergence.

## 5 Experiments

In this section, we conduct extensive experiments on three widely-used CRS benchmarks to study the following research questions: (1) Can CRSIRL achieve better performance than state-of-the-art CRS solutions? (2) How does each proposed component contribute to the final performance of CRSIRL? (3) How does the degree of intrinsic rewards affect the policy learning?

**Algorithm 1:** Optimization algorithm of CRSIRL

Initialize $\theta$ and $\phi$;
Initialize the trajectory buffer $\mathcal{B} \leftarrow \varnothing$;
**while** *not* Done **do**

    `# Inner Loop`
    Collect $\tau_i$ by executing $\pi_\theta$;
    Compute updated parameters $\theta'$ with intrinsic reward $r^{ex} + \lambda r_\phi^{in}$ using Eq.(9);
    `# Outer Loop`
    Collect $\tau_o$ by executing $\pi_{\theta'}$;
    Compute $\mathcal{L}^{ex}$ with shaped reward $\tilde{R}^{ex}$ using Eq.(5) ;
    Sample $\tau^0, \tau^1 \sim \mathcal{B}$; Compute $\mathcal{L}^p$ using Eq.(7);
    Update $\phi$ with gradients computed by Eq.(13);

**end**

## 5.1 Setup

**Datasets.** We evaluate CRSIRL on three multi-round CRS benchmarks [Lei et al., 2020a, Deng et al., 2021]. The **LastFM** dataset is for music artist recommendation. Lei et al. [2020a] manually grouped the original attributes into 33 coarse-grained attributes. The **LastFM\*** dataset is the version where attributes are not grouped. The **Yelp\*** dataset is for local business recommendation. We summarize their statistics in Table 1.

**User simulator.** Training and evaluating CRS through direct user interactions can be prohibitively expensive at scale. We address this by employing the user simulator approach from [Lei et al., 2020a], simulating a conversation session for each observed user-item interaction pair $(u, v)$. In this simulation, item $v$ is considered the target item, and its attribute set $\mathcal{P}_v$ is treated as the oracle set of attributes preferred by user $u$. The session begins with the simulated user specifying an attribute, randomly selected from $\mathcal{P}_v$. This simulation adheres to the "System Ask, User Respond" paradigm in CRS, as described in [Zhang et al., 2018].

Table 1: Summary statistics of datasets.

|  | LastFM | LastFM* | Yelp* |
|---|---|---|---|
| #Users | 1,801 | 1,801 | 27,675 |
| #Items | 7,432 | 7,432 | 70,311 |
| #Attributes | 33 | 8,438 | 590 |
| #Interactions | 76,693 | 76,693 | 1,368,606 |

**Baselines.** We consider a rich set of state-of-the-art CRS solutions. **Max Entropy** chooses to select an attribute with maximum entropy based on the current state, or to recommend the top ranked item. **Abs Greedy** [Christakopoulou et al., 2016] continues to suggest items until it either makes a successful recommendation or reaches the maximum number of allowed attempts. **CRM** [Sun and Zhang, 2018] is an RL-based solution. It integrates user preferences into a belief tracker, which then guides the decision-making process regarding when to ask which attribute. **EAR** [Lei et al., 2020a] proposes a three-stage RL solution consisting of estimation, action and reflection. **SCPR** [Lei et al., 2020b] reconceptualizes the CRS problem as an interactive path reasoning process within a user-item-attribute graph. It selects candidate attributes and items based on their relationship to attributes that have already interacted with users within this graph. **FPAN** [Xu et al., 2021] extends the EAR model by utilizing a user-item-attribute graph to enhance offline representation learning. User embeddings are revised dynamically based on users' feedback on items and attributes in the conversation. **UNICORN** [Deng et al., 2021] merges the conversation and recommendation components into a unified RL agent. To streamline the RL training process, it proposes two heuristic strategies for pre-selecting attributes and items at each turn.

**Evaluation metrics.** We follow previous works on multi-round CRS to evaluate the performance of CRS with success rate at turn $T$ (SR@$T$) and average turns (AT) of conversations. SR@$T$ is the average ratio of successful episodes with $T$ turns, while AT is the average number of turns for all conversations. We also report the two-level hierarchical normalized discounted cumulative gain [Deng et al., 2021] defined as

$$hDCG@(T, K) = \sum_{t=1}^{T} \sum_{k=1}^{K} r(t, k) \left[ \frac{1}{\log_2(t+2)} + \left( \frac{1}{\log_2(t+1)} - \frac{1}{\log_2(t+2)} \right) \frac{1}{\log_2(k+1)} \right],$$

Table 2: Main results. For SR@15 and hDCG, higher is better. For AT, lower is better. $^\dagger$ represents the improvement over all baselines is statistically significant with $p$-value $< 0.01$.

| | LastFM | | | LastFM* | | | Yelp* | | |
|---|---|---|---|---|---|---|---|---|---|
| | SR@15 | AT | hDCG | SR@15 | AT | hDCG | SR@15 | AT | hDCG |
| Abs Greedy | 0.222 | 13.48 | 0.073 | 0.635 | 8.66 | 0.267 | 0.189 | 13.43 | 0.089 |
| Max Entropy | 0.283 | 13.91 | 0.083 | 0.669 | 9.33 | 0.269 | 0.398 | 13.42 | 0.121 |
| CRM | 0.325 | 13.75 | 0.092 | 0.580 | 10.79 | 0.224 | 0.177 | 13.69 | 0.070 |
| EAR | 0.429 | 12.88 | 0.136 | 0.595 | 10.51 | 0.230 | 0.182 | 13.63 | 0.079 |
| SCPR | 0.465 | 12.86 | 0.139 | 0.709 | 8.43 | 0.317 | 0.489 | 12.62 | 0.159 |
| FPAN | 0.630 | 10.16 | 0.224 | 0.667 | 7.82 | 0.407 | 0.236 | 12.77 | 0.116 |
| UNICORN | 0.535 | 11.82 | 0.175 | 0.788 | 7.58 | 0.349 | 0.520 | 11.31 | 0.203 |
| **CRSIRL** | **0.772**$^\dagger$ | **10.12**$^\dagger$ | **0.231**$^\dagger$ | **0.913**$^\dagger$ | **6.79**$^\dagger$ | **0.431**$^\dagger$ | **0.622**$^\dagger$ | **10.61**$^\dagger$ | **0.228**$^\dagger$ |

Table 3: Abalation study of different components of CRSIRL.

| | LastFM | | | LastFM* | | | Yelp* | | |
|---|---|---|---|---|---|---|---|---|---|
| | SR@15 | AT | hDCG | SR@15 | AT | hDCG | SR@15 | AT | hDCG |
| PG | 0.724 | 10.42 | 0.217 | 0.882 | 7.41 | 0.401 | 0.598 | 10.95 | 0.186 |
| ¬HRS | 0.732 | 10.58 | 0.219 | 0.898 | 7.16 | 0.401 | 0.602 | 11.21 | 0.196 |
| ¬RPM | 0.754 | 10.14 | 0.227 | 0.904 | 6.89 | 0.415 | 0.606 | 10.81 | 0.213 |
| MTL | 0.768 | **10.09** | 0.224 | 0.908 | 6.92 | 0.426 | 0.613 | 10.73 | 0.207 |
| **CRSIRL** | **0.772** | 10.12 | **0.231** | **0.913** | **6.79** | **0.431** | **0.622** | **10.61** | **0.228** |

where $T$ and $K$ represent the number of conversation turns and recommended items in each turn, $r(t, k)$ denotes the relevance of the results at the turn $t$ and position $k$. Intuitively, successful sessions with fewer turns are preferable for CRS. Also, the target item is expected to be ranked higher on the recommendation list at the success turn. We report $hDCG@(15, 10)$ by default.

**Training details.** All datasets are split by 7:1.5:1.5 ratio for training, validation and testing. We used the Transformer-based state encoder proposed in [Chu et al., 2023]. We adopt TransE [Bordes et al., 2013] to pretrain the node embeddings on the training set, and use the user simulator described before for online policy learning on the validation set. We first pretrain the policy with only extrinsic reward using policy gradient and then apply CRSIRL to fine-tune the pretrained policy. The learning rates in the inner and outer loop are searched from $\{1e^{-5}, 5e^{-5}, 1e^{-4}\}$ with Adam optimizer. The coefficient of intrinsic reward $\lambda$ is searched from $\{0.05, 0.1, 0.5, 1.0\}$. The discount factor $\gamma$ is set to 0.999. All experiments are run on an NVIDIA Geforce RTX 3080Ti GPU with 12 GB memory. RL-based baselines rely on handcrafted rewards, we follow Lei et al. [2020a] to set (1) $r_{\text{rec\_suc}} = 1$ for successful recommendation; (2) $r_{\text{rec\_fail}} = -0.1$ for failed recommendation; (3) $r_{\text{ask\_suc}} = 0.1$ when the inquired attribute is confirmed by the user; (4) $r_{\text{rec\_fail}} = -0.1$ when the inquired attribute is dismissed by the user; (5) $r_{\text{quit}} = -0.3$ when the user quits the conversation without a successful recommendation. We set the maximum turn $T$ as 15 and the size $K$ of the recommendation list as 10. We provide more implementation details in the supplementary material.

## 5.2 Results & Analysis

We present the main results in Table 2. We can clearly observe the CRSIRL outperformed all baselines with a large margin. Both FPAN and EAR are policy gradient based methods, but they pretrain their policies using conversation history generated by a rule-based strategy via supervised learning. This training approach biases policies towards pre-set rules, limiting the performance of policy learning on datasets with larger action spaces (like LastFM* and Yelp*), where more exploration is necessary. SCPR and UNICORN have relatively stable performance on all the datasets. Our CRSIRL outperforms all baselines significantly with its learned intrinsic rewards. Rather than arbitrarily assigning the reward values, we dynamically optimize them in CRSIRL. Any action that contributes to a final successful recommendation should receive credit and thus be promoted by policy learning, regardless of whether it involves a rejected attribute or a failed recommendation.

## 5.3 Ablation Study

**Contributions of each component in CRSIRL.** We evaluate different variants of CRSIRL to study the contributions of each proposed component. Firstly, we disable the fine-tuning with CRSIRL and directly report the results after the policy gradient pretraining with only extrinsic rewards, denoted as PG. Secondly, we remove HRS and RPM to evaluate their individual effectiveness. In both variants, the outer loop degenerates to a single objective optimization problem. Finally, we conduct an experiment where instead of updating the objectives with MOO, we treat the two objectives as distinct tasks and assign them equal weights, a process referred to as Multi-Task Learning (MTL).

We present the results in Table 3. Interestingly, we observe that directly optimizing the extrinsic rewards with policy gradient already outperformed most of baselines. It is worth noting that PG uses sparser rewards than other baselines in Table 2 with manually-defined rewards for intermediate actions. However, the exploratory behavior of PG enables it to outperform these baselines. We can observe that without HRS, the AT metric degenerated on all three datasets. HRS prefers actions which can increase the rank of the target item, which is the most direct metric of action utilities in CRS. Even though the asked questions could still be helpful without HRS, HRS provides explicit hints about how to ask the most useful questions, leading to a smaller AT. Besides, the performance decreases after removing RPM, which finds actions leading to successful recommendations by comparing successful and failed trajectories. Lastly, MTL shows a significant improvement compared to PG, and it occasionally outperforms CRSIRL (e.g., AT on LastFM). However, MTL has difficulty balancing the two objectives, generally resulting in worse performance than CRSIRL.

**Analysis of $\lambda$.** We use the hyper-parameter $\lambda$ to control the influence of intrinsic rewards. It is important to study how it affects the policy learning. In this experiment, we evaluate CRSIRL with $\lambda = \{0.05, 0.1, 0.5, 1.0\}$ on the LastFM dataset. The result is shown in Figure 3. We can clearly observe that the performance peaks when an appropriate $\lambda$ is chosen. With a small $\lambda$, the intrinsic reward is not strong enough to affect the final performance. However, a large value of $\lambda$ can unfortunately impair performance. This is because the estimation errors in the intrinsic reward could overwhelm the extrinsic reward. Even though it is sparse, the extrinsic reward can help calibrate the intrinsic reward.

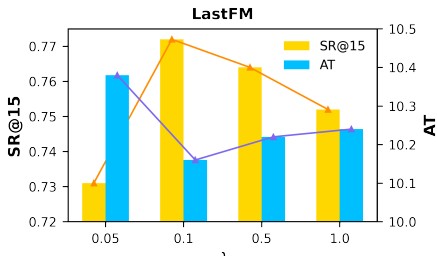

Figure 3: Performance with different $\lambda$.

## 5.4 Case Study

Additionally, we performed a qualitative study to analyze the learned intrinsic rewards of CRSIRL (shown in Figure 4) on the LastFM dataset. The natural language questions and user responses are generated by predefined templates. We observe that the intrinsic rewards depend not only on whether the user accepts or rejects the action, but also on how well the action contributes to the final recommendation. Even though the user accepts *pop*, the intrinsic reward for this action remains negative. This is because *pop* is a very general attribute and contributes little to modeling the user's preference. Conversely, although *vocalist* is rejected by the user, it still carries a small positive value as it aids in identifying the target artist. Finally, *Indie* and *Punk* are two attributes that are accepted and best describe the target artist, *Franz Ferdinand*[2] (a band known for *indie rock* and *post-punk revival*). Consequently, they carry relatively large positive intrinsic rewards. This case shows the CRSIRL can provide more fine-grained reward signals, leading to better final performance.

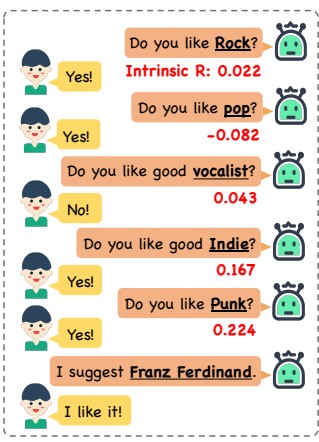

Figure 4: Conversations generated by CRSIRL. The values of the learned intrinsic rewards are marked in red.

---

[2] https://en.wikipedia.org/wiki/Franz_Ferdinand_(band)

# 6 Conclusions

In this paper, we study an important but largely under-explored problem in CRS, i.e., reward function design. We present a principled solution for reward learning for CRS and formulate an online algorithm to learn intrinsic rewards via bi-level multi-objective optimization. In the inner loop of CRSIRL, we optimize the policy with the learned intrinsic reward function. And in the outer loop, we optimize the intrinsic reward function to satisfy two criteria designed for CRS: maximizing success rate and minimizing number of turns in conversations. The results on three CRS benchmarks demonstrated the effectiveness of learned intrinsic rewards.

CRSIRL sheds light on learning reward functions to improve CRS. Currently, we consider two directly quantifiable objectives for CRS, i.e., success rate and conversation length. Other perspectives, such as user satisfaction [Liang et al., 2006] and fairness [Lin et al., 2022], are worth to be investigated and embedded into reward learning. Moreover, beyond the template-based conversation generation, it is important to integrate CRSIRL with advanced natural language-based conversational agents, such as [Touvron et al., 2023, Zhang et al., 2023], to learn reward functions that satisfy multiple objectives favored by humans during natural language driven interactions, such as conversational persuasiveness, cohesiveness and explainability [Moon et al., 2019].

# 7 Acknowledgement

We thank the anonymous reviewers for their insightful comments. This work was partially supported by NSF IIS-2007492, IIS-2128019 and NSF IIS-1838615.

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
