# OpenReview forum: "Multi-Objective Intrinsic Reward Learning for Conversational Recommender Systems"
_NeurIPS.cc/2023/Conference — NeurIPS 2023 poster_

### Official Review · Reviewer_5ZDc · 2023-07-04

**Soundness:** 3 good
**Presentation:** 3 good
**Contribution:** 2 fair
**Rating:** 5
**Confidence:** 3

**Summary:**

In this paper, the authors propose an algorithm for learning the intrinsic reward function in order to solve the problem of poor results due to improper design of the reward function of the dialogue strategy module in current conversation recommendation systems. Specifically, a multi-objective bi-level optimization problem is designed, where the inner layer uses the learned reward function to optimize the selection strategy, and the outer layer updates the reward function used to optimize the system metrics. Excellent results are achieved in two conversation recommendation datasets.



**Strengths:**

It is interesting research direction to design an internal reward based on the results of each interaction, in addition to the external reward generated by the conversation results, to satisfy the results of two recommendation goals through a multi-objective bi-level optimization framework.
A hindsight internal reward function was designed to calculate the reward score of the current strategy after each target item was hit.
In order to obtain successful recommendation results faster and thus reduce the number of conversation rounds, the authors propose the Recommendation Preference Matching module, which improves the possibility of selecting the right decision.
The authors designed a Multi-Objective Bi-Level Optimization strategy in which the inner loop, using two rewards, optimizes the decisions of the conversation, while the outer loop optimizes the reward function.
The experiment design is comprehensive and solid, and the result analysis is clear and convincing.


**Weaknesses:**

First of all, your baseline is not enough, the CRIF model published in SIGIR in 2022 is not compared, and from the results of CRIF, your performance is worse than it, which needs to be verified by adding more experiments.
I can't understand the example you showed in Case Study, pop and rock are very common examples, and Franz Ferdinand also has the label of pop rock, why pop gave a negative score it, and the contribution of this article is mainly in the conversation policy module, in the second step to raise a completely useless question, is this a policy failure?
Your work contains only the content of the dialogue policy module, and in the experimental part we can see that it has good recommendation performance, but the article does not describe how to get the recommendation results, please explain.
It is suggested to include a significance test to prove the validity of the experiment.

**Questions:**

Please refer to weakness.

**Limitations:**

Please refer to weakness.

---

> ### Author Rebuttal · Authors · 2023-08-10
>
> We thank the reviewer for the constructive suggestions to strengthen the empirical support of our work by adding more advanced baselines and significance testing.
>
> ---
>
> **[Q1]** Compare with CRIF
>
> **[A1]** We discussed the CRIF model in our appendix and reported the experiment comparisons with it in the appendix B. Please kindly check it in the supplementary material. CRIF depends on a heuristic to identify the best action at each round of interaction. However, this heuristic abuses the design of the user simulator that all attributes of the target item will be accepted. This creates a form of information leakage and an unfair advantage over other solutions that do not exploit this specific knowledge. Thus we did not include it in the main paper.
>
> However, it is still interesting to understand whether the learned intrinsic rewards can augment this strong heuristic. The experiment results in appendix B showed that learning intrinsic rewards directly from user interactions offers an effective augmentation to CRIF, resulting in better performance. We will explicitly refer to this study and its conclusions in the main paper.
>
> ---
>
> **[Q2]** Recommendation results
>
> **[A2]** Building upon [1], we employ a unified policy in which the action space consists of both attributes and items. When the policy selects an attribute action, the agent inquires if the user prefers that specific attribute. Conversely, if an item action is chosen, the agent compiles a recommendation list for the user, with items ranked based on the scores assigned by the policy. A better policy is recognized by its ability to not only strategize questions more effectively (i.e., the average turn metric) but also produce an optimized recommendation list (i.e., the ranking quality). To evaluate the quality of the recommendation list, we utilize the hDCG metric. We will add a detailed description of how to obtain the recommendation results in the final version.
>
> [1] Yang Deng, Yaliang Li, Fei Sun, Bolin Ding, and Wai Lam. Unified conversational recommendation policy learning via graph-based reinforcement learning. arXiv preprint arXiv:2105.09710, 2021.
>
> ---
>
> **[Q3]** Clarification of the Case Study
>
> **[A3]** Thank you for your insightful observation regarding the Case Study! In the provided example, even though "pop" was accepted by the user, it received a negative intrinsic reward because it is a broad genre, associated with too many artists. This means that while "pop" may be liked, it is still not the most informative for refining recommendations at the moment. As a result, the intrinsic reward function gives "pop" a slightly negative reward, despite Franz Ferdinand falling under "pop rock". While the asked questions may occasionally seem redundant, it’s expected as the policy tries to avoid jumping to conclusions based on limited information (e.g., recommending at the beginning of the conversation). We see it as laying a foundational understanding of a user's preferences.
>
> ---
>
> **[Q4]** Significance test
>
> **[A4]** Thanks for your suggestion! All the experiments are repeated 3 times with different random seeds, and we report the final averaged metrics. We will add the results of the significance test to ensure the validity of the results.

---

### Official Review · Reviewer_vRxN · 2023-07-05

**Soundness:** 2 fair
**Presentation:** 2 fair
**Contribution:** 2 fair
**Rating:** 5
**Confidence:** 4

**Summary:**

This paper study the problem of reinforcement learning based conversational recommender systems. The paper claims that it is difficult to design a handcraft reward function for each step of the conversations. Thus the paper proposes a multi-objective bi-level optimization method that the inner level optimizes the policy with real rewards and the learned intrinsic rewards, and the outer level optimizes two CRS objectives: maximizing the success rate and minimizing the number of turns in successful conversations. The paper validates the effectiveness of the proposed methods in 3 CRS datasets.

**Strengths:**

1.The motivation of the paper is very clear.
2.The idea of hindsight reward shaping is sound and interesting.
3.The multi-objective bi-level optimization is also interesting and easy to follow.
4.The paper provide detailed introductions of evaluation metrics, and the performance improvement is significant.
5.The paper shows the ablation study of the HRS and the RPM component.

**Weaknesses:**

1.The major concern of the paper is that it is not clear that how the proposed techniques work together. First of all, the intrinsic reward function defined in Eq.(3) is related to the rank of the target item under the state. How is it related to the parameter \phi?

2.How does the RPM module work? It seems we train the policy with Eq.(7). Is it related to the intrinsic reward?

3.The performance improvement of CRSIRL is marginal compared with MTL, which means that we do need to use the MDGA algorithm.

**Questions:**

See the weakness.

**Limitations:**

See the weakness.

---

> ### Author Rebuttal · Authors · 2023-08-10
>
> We thank the reviewer for recognizing the motivation of our paper and our idea being interesting. We hope the following responses can address the reviewer’s concerns.
>
> ---
>
> **[Q1]** Clarification about how the components of CRSIRL work together
>
> **[A1]** We develop a bi-level optimization framework to learn intrinsic rewards from users’ explicitly provided reward signals for CRS. Our solution consists of an inner loop and an outer loop for the induced bi-level optimization problem. In the inner loop, we update the policy $\pi$ with the learned intrinsic reward (parameterized by $\phi$) to obtain an updated policy $\pi’$. In the outer loop, we update $\phi$ by calculating the meta loss $\mathbb{L}$ induced by $\pi’$ over the users’ provided explicit reward signal. Intuitively, we hope the learned intrinsic rewards can help us find a better policy $\pi’$, leading to a lower meta-loss.
>
> More specifically, to obtain the gradient of $\phi$, in the inner loop, we create a differentiable connection between $\pi$ and $\phi$ via gradient update (Eq.9). Such a connection allows us to calculate the gradient of $\phi$ w.r.t. the meta loss using chain rule (Eq.10 and 13). After obtaining the gradient of $\phi$ in the outer loop, we are able to update it using Eq.14.
>
> To better leverage explicit reward from user feedback, we design two CRS-specific objectives HRS (Eq.3) and RPM (Eq.7), and construct the multi-objective meta loss $\mathbb{L}$. Hence, we will not directly use HRS and RPM to update the policy, but use them to guide the learning of intrinsic reward. We will add the pseudo code of CRSIRL to facilitate better understanding in the final version.
>
> ---
>
> **[Q2]** Performance comparison between CRSIRL and MTL
>
> **[A2]** While MTL is a strong baseline that leverages our proposed HRS and RPM objectives, its static weight setting between the two objectives makes it less adaptable in different environments. This rigidity can result in one objective dominating the other. For example, in Table 3, MTL is skewed towards optimizing the average turn on the LastFM dataset.
>
> In contrast, CRSIRL does not merely establish a simple trade-off between the two objectives by a manually-set hyper-parameter, but strives to optimize them concurrently (as explained between line 221-234). By utilizing the MDGA algorithm, CRSIRL can flexibly adjust the weights between objectives, striving for a Pareto optimal solution, as supported by the findings in [1]. This ensures simultaneous optimization of both objectives without one excessively dominating the other. Our multi-objective framework not only theoretically avoids suboptimal outcomes but also demonstrates better empirical results over MTL, as observed in Table 3.
>
> Furthermore, our experiments were repeated three times with different random seeds, and the improvement brought by CRSIRL over MTL has been found to be statistically significant ($p<0.05$). We'll be incorporating the significance test results in the final version of the paper.
>
> [1] Désidéri, Jean-Antoine. "Multiple-gradient descent algorithm (MGDA) for multiobjective optimization." Comptes Rendus Mathematique 350.5-6 (2012): 313-318.

---

> > ### Comment · Reviewer_vRxN · 2023-08-16
> > **Reply**
> >
> > Thanks for the rebuttal, I am now pleased with the paper and I have increased the score.

---

### Official Review · Reviewer_Kjom · 2023-07-07

**Soundness:** 3 good
**Presentation:** 3 good
**Contribution:** 2 fair
**Rating:** 5
**Confidence:** 3

**Summary:**

Mainstream reinforcement learning-based CRS solutions heavily rely on handcrafted reward functions, which may not be aligned with user intent in CRS tasks. Therefore, the design of task-specific rewards is critical to facilitate CRS policy learning, which remains largely under-explored in the literature. This paper proposes a novel approach to address this challenge by learning intrinsic rewards from interactions with users. Specifically, the paper formulates intrinsic reward learning as a multi-objective bi-level optimization problem. Extensive experiments on three public CRS benchmarks show that the developed algorithm significantly improves CRS performance by exploiting informative learned intrinsic rewards.

**Strengths:**

(1) The paper is well motivated, with a convincing motivating example.
(2) The develop technique is solid and has clear intuitions.
(3) Clear improvements on multiple datasets are observed in the experiments.

**Weaknesses:**

(1) More experiment details can be included. Previous works usually show the curves reporting the performance with different training episodes and different turns of conversations, for example, in baselines

Yang Deng, Yaliang Li, Fei Sun, Bolin Ding, and Wai Lam. Unified conversational recommendation policy learning via graph-based reinforcement learning. arXiv preprint arXiv:2105.09710, 2021.

Wenqiang Lei, Xiangnan He, Yisong Miao, Qingyun Wu, Richang Hong, Min-Yen Kan, and Tat-Seng Chua. Estimation-action-reflection: Towards deep interaction between conversational and recommender systems. In Proceedings of the 13th WSDM Conference, pages 304–312, 2020a.

Usually learning a more generic reward function may need more samples compared to learning a handcrafted reward function, although after convergence, learning the generic reward function can lead to better performance compared to learning a handcrafted reward function. It would be interesting if the paper can report SR@K, where K is a small number, because the user experience in early interactions is very important in conversational recommendations.

(2) The way of user interactions in Figure 1 (System Ask, User Respond) may become less interesting, especially when nowadays where users can progressively and flexibly describe what they want by LLMs with decent accuracy of understanding the desired attributes. For example, as in Table 2, by all the algorithms in this setting (System Ask, User Respond), even after 15 interactions, very few algorithms can achieve satisfying recommendations. However, if the user can progressively and flexibly describe what they want by LLMs, it is likely that the user can get proper recommended items with only very few interactions. Even the user may not be progressive in some cases, a hybrid setting (where the user can either respond yes/no or progressively describe what they want during the conversations) is more practical.

**Questions:**

(1) What are the comparison results, using the metric SR@K, where K is a small number (e.g., 3 or 5)?

(2) Are the inner loop optimization and outer loop optimization in an alternating fashion during training? If so, is there any insight or theoretical analysis about the convergence of the training procedure?

**Limitations:**

Limitations seem not discussed in the paper.

---

> ### Author Rebuttal · Authors · 2023-08-10
>
> We appreciate the reviewer for the constructive comments on enriching experiment results and more practical CRS paradigm.
>
> ---
>
> **[Q1]** What are the comparison results, using the metric SR@K, where K is a small number (e.g., 3 or 5)?
>
> **[A1]** We report the SR@5 of CRSIRL, compared with the strongest baseline UNICORN,
>
> |         | LastFM | LastFM* | Yelp* |
> |:-------:|:------:|:-------:|:-----:|
> | UNICORN |  0.104 |  0.215  | 0.068 |
> |  CRSIRL |  0.262 |  0.324  | 0.182 |
>
> This notable performance gain can be attributed to CRSIRL's strategy of posing more informative questions early in the conversation, allowing it to make more accurate recommendations as the conversation progresses. We will include more experiment details in the final version.
>
> ---
>
> **[Q2]** Are the inner loop optimization and outer loop optimization in an alternating fashion during training? If so, is there any insight or theoretical analysis about the convergence of the training procedure?
>
> **[A2]** The inner and outer loops are operated in an alternative fashion, as all other solutions for bi-level optimization. Please refer to the general response CQ1 for the convergence analysis of CRSIRL.
>
> ---
>
> **[Q3]** Progressive CRS paradigm
>
> **[A3]** Progressive CRS presents a more reasonable paradigm, allowing users to take a more active role in the conversation by explicitly describing their preferences. The idea of learning CRS with such actively participative users is interesting. However, at the time of our paper's submission, creating a progressive simulator based on LLMs is still an emerging field of study [1]. It's important to clarify that designing such a progressive user simulator, or any specific type of user simulator, is not the focus of our paper. Instead, our primary objective is the effective learning of optimal reward functions for CRS. For evaluations, we employed the widely-accepted "System Ask, User Respond" [2] paradigm. While the fundamental mechanism of reward learning remains consistent, irrespective of the CRS setting, we believe that the integration of our proposed method within the progressive CRS paradigm will be a seamless process. Exploring this integration further is a promising direction and we leave it as an important future work.
>
> [1] Wang, Xiaolei, et al. "Rethinking the Evaluation for Conversational Recommendation in the Era of Large Language Models." arXiv preprint arXiv:2305.13112 (2023).
>
> [2] Zhang, Yongfeng, et al. "Towards conversational search and recommendation: System ask, user respond." Proceedings of the 27th acm international conference on information and knowledge management. 2018.

---

> > ### Comment · Reviewer_Kjom · 2023-08-21
> > **Thanks for the responses**
> >
> > Thanks for the responses. I acknowledge that I have read the responses and would keep the score.

---

### Official Review · Reviewer_reFg · 2023-07-10

**Soundness:** 3 good
**Presentation:** 3 good
**Contribution:** 2 fair
**Rating:** 6
**Confidence:** 3

**Summary:**

The paper addresses the problem of designing effective reward functions for conversational recommender systems (CRS), which is critical but largely under-explored in the literature. The paper proposes a novel approach to learn intrinsic rewards from user feedback, which can better capture the user intent and optimize multiple CRS-specific objectives, such as success rate and conversation length. The paper formulates intrinsic reward learning as a multi-objective bi-level optimization problem, and develops an algorithm to solve it. The paper evaluates the proposed approach on three public CRS benchmarks, and shows that it significantly improves CRS performance by exploiting informative learned intrinsic rewards.

**Strengths:**

1) Originality: The paper proposes a novel approach to learn intrinsic rewards from user feedback, which can better capture the user intent and optimize multiple CRS-specific objectives. This represents a creative combination of existing ideas and a new problem formulation in the field of conversational recommender systems.
2) Quality: The paper formulates intrinsic reward learning as a multi-objective bi-level optimization problem, and develops an online algorithm to solve it. The proposed approach is rigorously evaluated on three public CRS benchmarks, and shows significant improvement in CRS performance by exploiting informative learned intrinsic rewards.
3) Clarity: The paper is well-written and clearly presents the research problem, methodology, results, and conclusions. The technical details are explained in a clear and concise manner, making it accessible to a broad audience.
4) Significance: It is questionable if The paper makes a significant contribution to the field of conversational recommender systems. I am not sure if any of this will actually be used in practical systems because algorithms are evaluated on non practical scenarios/variants of datasets.

**Weaknesses:**

1) The paper does not provide any details on convergence of the algorithm or computations complexity of the proposed method against existing methods.
2) Even though it seems like a useful strategy to use intrinsic rewards, It is unclear how the intrinsic rewards affect the policy learning process. This is not well motivated using experimental results.
3) How does a user simulator affect overall results? Can authors justify the specific choice of user simulator? Why does the attribute set Pv be treated as the oracle set?
4) It is also unclear how the hyperparameters of the model are tuned and how they affect the performance.
5) Not exactly a weakness but this would be an interesting study too: The paper does not compare the proposed method with any baselines that use real user feedback, such as ratings, reviews, or explanations. It is possible that the user simulator may not capture the true user preferences or behavior, and that the intrinsic rewards may not reflect the real user satisfaction. It is also possible that the user feedback may provide additional information or guidance for improving the recommendation quality and user satisfaction.
6) The paper does not evaluate the proposed method on a diverse set of datasets such as e-commerce, music, or movies. It is unclear how the proposed method would generalize to different domains, contexts, or user types. It is also unclear how the proposed method would handle practical challenges, such as data sparsity, noise, or diversity.

**Questions:**

1) How does the user simulator approach compare to other methods of evaluating CRS solutions, such as human evaluation or simulation with real user logs?
2) How does the proposed CRSIRL handle the exploration-exploitation trade-off in policy learning, especially when the intrinsic rewards are uncertain or noisy?
3) How does the multi-objective bi-level optimization framework deal with the potential conflicts or trade-offs between the two objectives of maximizing success rate and minimizing number of turns? Authors mention this in section 3 but it's not analyzed in the results section.
4) Can CRSIRL incorporate user feedback on the generated recommendations, such as ratings, reviews, or explanations?
5) Can CRSIRL handle the cold start problem, when there is no or little prior information about the user preferences or behavior?
6) Can CRSIRL cope with the dynamic and evolving nature of the user preferences and behavior, especially in long-term interactions?
7) How does the CRSIRL leverage the contextual information, such as user profile, location, time, or mood, to improve the recommendation quality and user satisfaction?

---

> ### Author Rebuttal · Authors · 2023-08-10
>
> We thank the reviewer for the positive comments on our work, and the constructive questions to help enrich our work in the future.
>
> ---
>
> **[Q1]** How does the user simulator approach compare to other methods of evaluating CRS solutions?
>
> **[A1]** Due to interactive nature of conversational recommendation, training and evaluating CRS with real users is costly. We use a user simulator, inspired by [1], for efficient training and evaluation. Assuming users favor all attributes of the target item, we treat the attribute set $P_v$ as the oracle, ensuring consistent feedback. While crafting a more realistic simulator is compelling, our paper focuses on reward learning. For fair comparisons, we choose to align our evaluation with established baselines
>
>
> On the other hand, using logged data to learn or evaluate a RL algorithm is known to be challenging, due to the issue of distribution shift. Though various solutions have been proposed, they have different limitations (e.g., variance vs., bias trade-off, and computational complexity), which add additional complexity in our study. Given we are the first to study the reward learning problem in CRS, we choose simulations to evaluate our proposed solution.
>
>
> [1] Sun, Yueming, and Yi Zhang. "Conversational recommender system." The 41st international acm sigir conference on research & development in information retrieval. 2018.
>
> ---
>
> **[Q2]** How does the proposed CRSIRL handle the exploration-exploitation trade-off in policy learning?
>
> **[A2]** While intrinsic rewards can be noisy, our framework relies on noise-free extrinsic rewards in the outer loop for stable feedback. These extrinsic rewards offer a clear training signal, enabling CRSIRL to refine intrinsic rewards and guide the policy to more effective actions. As a result, despite the noisy intrinsic feedback, CRSIRL maintains a balance between exploration and exploitation during the training. Additionally, as suggested by [1], noisy feedback can enhance exploration, potentially benefiting early training stages.
>
> [1] Kannan, Sampath, et al. "A smoothed analysis of the greedy algorithm for the linear contextual bandit problem." Advances in neural information processing systems 31 (2018).
>
> ---
>
> **[Q3]** Potential conflicts or trade-offs between the two objectives
>
> **[A3]** Our multi-objective optimization framework adeptly navigates these trade-offs, ensuring convergence to a point on the Pareto frontier, as evidenced in [1]. CRSIRL does not merely establish an arbitrary trade-off between the two objectives by a manually-set hyper-parameter, but strives to optimize them concurrently (as explained between line 221-234). CRSIRL realizes it by dynamically adjusting the weights, denoted as $\alpha$, for each objective throughout the training process. As Table 3 illustrates, while the multi-task learning variant can indeed benefit from the dual objectives, CRSIRL can still outperform it by establishing a more desirable trade-off between the two objectives.
>
> [1] Désidéri, Jean-Antoine. "Multiple-gradient descent algorithm (MGDA) for multiobjective optimization." Comptes Rendus Mathematique 350.5-6 (2012): 313-318.
>
> ---
>
> **[Q4]** Can CRSIRL incorporate user feedback on the generated recommendations?
>
> **[A4]** CRSIRL is capable of integrating various types of user feedback, such as ratings, reviews, or explanations, as signals of extrinsic reward. And they can be included as other optimization objectives within our multi-objective framework. However, it's worth noting that incorporating user feedback would not fundamentally alter the focus of this paper, which is to investigate effective reward learning strategies in CRS. We leave how to effectively utilize other types of user feedback for intrinsic reward learning as an important future work.
>
> ---
>
> **[Q5]** Can CRSIRL handle the cold start problem, when there is no or little prior information about the user preferences or behavior?
>
> **[A5]** CRS is expected to handle cold-start problems in recommendation by profiling a new user via eliciting her preference about item attributes on the fly. And CRSIRL is expected to better handle the cold-start problem by learning more effective rewards to guide CRS policy learning.
>
> ---
>
> **[Q6]** Can CRSIRL cope with the dynamic and evolving nature of the user preferences and behavior, especially in long-term interactions?
>
> **[A6]** The dynamic and evolving nature of user preferences actually signal a shift in the reward distribution, which poses a non-stationary learning environment. This is known as a significant challenge in reinforcement learning in general. Not only does the learned reward function but also the policy learning algorithm has to cope with it. Currently, the CRSIRL framework is not designed to handle such a non-stationary environment, particularly in the context of long-term interactions where user behavior and preferences may evolve considerably. The reviewer’s comment is very well taken, and we acknowledge this as an area for potential enhancement of our framework.
>
> ---
>
> **[Q7]** How does the CRSIRL leverage the contextual information to improve the recommendation quality and user satisfaction?
>
> **[A7]** Currently, we learn the intrinsic reward function to maximize the recommendation quality. However, it is possible to further improve user satisfaction by learning context-aware reward functions by encoding contextual information into reward functions, i.e., context-aware CRS. For example, different context induces different conversation strategies. It would be an interesting research direction for us in the future.
>
> ---
>
> **[Q8]** Hyper-parameter setting
>
> **[A8]** We provide the search ranges for hyper-parameters in Section 5.1 under "Training Details." Meanwhile, an in-depth discussion regarding the impact of different $\lambda$ is included in Section 5.3.
>
> ---
>
> **[Q9]** Convergence analysis
>
> **[A9]** Please refer to the general response CQ1.

---

> > ### Comment · Reviewer_reFg · 2023-08-21
> > **Thank you for the rebuttal**
> >
> > Thank you for the detailed rebuttal. Many of my questions were answered. The impact of convergence analysis was not clear and authors should connect that to experiments too. In any case, I am improving my score to 6.

---

### Author Rebuttal · Authors · 2023-08-10

We sincerely thank all the reviewers for their thoughtful comments and constructive suggestions, which will help us strengthen our paper. We are encouraged to find that the reviewers appreciate the clear presentation (reviewer reFg, vRxN), motivation of our study (reviewer reFg, Kjom, vRxN, 5ZDc), novelty of our approach (reviewer reFg, Kjom, vRxN), solid experiment design and improvement (reviewer reFg, Kjom, vRxN, 5ZDc). In the following, we will first provide the answer to the common question regarding the convergence analysis of our method, and then endeavor to provide individual responses to each reviewer.

---

**[CQ1]** Convergence analysis of CRSIRL (Reviewer reFg and Kjom)

**[CA1]** We provide the proof sketch of the convergence rate as follows.

Based on the results in [1], to prove the convergence of CRSIRL, we only need to prove our multi-objective meta loss function is Lipschitz smooth. Formally, we prove the following lemma,

*Lemma 1*. Given Lipschitz smooth loss functions $f$ and $g$, the following function is also Lipschitz smooth,

$\alpha f+(1-\alpha) g,$

where $\alpha \in [0, 1]$.

*Proof*. Let $f$ and $g$ be Lipschitz smooth with Lipschitz constants $L_f$ and $L_g$ respectively. This means:

1. $|f(x)-f(y)| \leq L_f|| x-y||$,  for all $(x, y)$,
2. $|g(x)-g(y)| \leq L_g || x-y||$, for all $(x, y)$.

Consider a function $h(x)=\alpha f(x) + (1-\alpha )g(x)$, we need to show $h(x)$ is Lipschitz smooth.

$
|h(x)-h(y)|=|\alpha f(x)+(1-\alpha)g(x)-\alpha f(y)-(1-\alpha)g(y)|
               =|\alpha (f(x)-f(y))+(1-\alpha )(g(x)-g(y))|.
$

Using the triangle inequality, we have

$
|h(x)-h(y)| \leq \alpha |(f(x)-f(y))|+|(1-\alpha)(g(x)-g(y))|
                       = \alpha |(f(x)-f(y))|+(1-\alpha)|(g(x)-g(y))|.
$

By using the inequalities we induced by Lipschitz smooth, we get

$
|h(x)-h(y)| \leq \alpha L_f || x-y ||+(1-\alpha)L_g||x-y||
           = (\alpha L_f +(1-\alpha)L_g)||x-y||.
$

Thus, the function $h$ is Lipschitz smooth with Lipschitz constant $L_h=\alpha L_f +(1-\alpha)L_g$.

*QED.*

By plugging *Lemma 1* into *Theorem 1* and *Theorem 2* of [1], we are able to prove the convergence guarantee of CRSIRL. We will provide the complete proof in the final version of our paper.

[1] Liu, Runze, et al. "Meta-reward-net: Implicitly differentiable reward learning for preference-based reinforcement learning." Advances in Neural Information Processing Systems 35 (2022): 22270-22284.

---

### Author Response · Authors · 2023-08-16
**A gentle reminder to the reviewers**

Dear reviewers,

Thank you again for your valuable time and thoughtful comments! We have provided thorough responses and additional results in response to your comments.

As we are approaching the end of the discussion stage, we would appreciate it if you could read our responses and update the scores if your concerns have been addressed. We are more than happy to further discuss any concerns that you find not fully addressed before the discussion period officially ends. Thank you!

Best regards,

Authors

---

### Decision · Program_Chairs · 2023-09-21

**Decision:**

Accept (poster)

**Comment:**

The paper studies multi-objective intrinsic reward learning to improve conversational recommender systems with multi-dimensional goals, such as maximizing success rate and shorten conversation length. Reviewers are overall happy with the studied problem, the proposed method and clarity of the presentation. The ablation study offer some clarity into the importance of the different components introduced, but the discussion on these results are short and do not offer much more insights.